# A Review: Non-Contact and Full-Field Strain Mapping Methods for Experimental Mechanics and Structural Health Monitoring

**DOI:** 10.3390/s24206573

**Published:** 2024-10-12

**Authors:** Wei Meng, Sergei M. Bachilo, R. Bruce Weisman, Satish Nagarajaiah

**Affiliations:** 1Department of Civil and Environmental Engineering, Rice University, Houston, TX 77005, USA; wm14@rice.edu; 2Department of Chemistry, Rice University, Houston, TX 77005, USA; bachilo@rice.edu (S.M.B.); weisman@rice.edu (R.B.W.); 3Department of Materials Science and NanoEngineering, Rice University, Houston, TX 77005, USA; 4Department of Mechanical Engineering, Rice University, Houston, TX 77005, USA

**Keywords:** strain measurement, strain mapping, experimental mechanics, structural health monitoring

## Abstract

Non-contact and full-field strain mapping captures strain across an entire surface, providing a complete two-dimensional (2D) strain distribution without attachment to sensors. It is an essential technique with wide-ranging applications across various industries, significantly contributing to experimental mechanics and structural health monitoring. Although there have been reviews that focus on specific methods, such as interferometric techniques or carbon nanotube-based strain sensors, a comprehensive comparison that evaluates these diverse methods together is lacking. This paper addresses this gap by focusing on strain mapping techniques specifically used in experimental mechanics and structural health monitoring. The fundamental principles of each method are illustrated with specific applications. Their performance characteristics are compared and analyzed to highlight strengths and limitations. The review concludes by discussing future challenges in strain mapping, providing insights into potential advancements and developments in this critical field.

## 1. Introduction

Mechanical strain is the deformation of a material as a result of stress. Strain measurement is crucial for understanding how materials and structures respond to external and internal forces. By measuring strain, engineers and researchers can assess the performance and durability of materials under various load conditions, predict and prevent potential failures by identifying areas of high stress, validate and refine computational models, collect real-life data to optimize maintenance, and ensure safety and reliability by monitoring structural integrity over time.

Strain measurement has widespread applications across various fields. In civil engineering, continuous strain monitoring of bridges, buildings, and dams helps detect early signs of stress and potential failure, ensuring structural integrity and safety [1,2,3]. Aerospace engineering relies on strain measurement for optimizing the design of aircraft wings, fuselages, spacecraft, and helicopter turbine blades to withstand extreme conditions and enhance safety standards [4,5,6]. Mechanical engineering uses strain data to improve the design and durability of automotive components, manufacturing equipment, and robotics, leading to safer and more efficient systems [7,8,9,10,11,12]. In bioengineering, strain measurement is crucial for understanding the mechanical properties of biological tissues and optimizing the design of orthopedic implants, cardiovascular devices, and engineered tissues [13,14,15]. The energy industry also benefits from strain measurement, which is used to monitor the structural health of wind turbine blades, pipelines, and offshore platforms, ensuring reliable efficient energy production and storage [16,17,18,19,20,21]. In the battery industry, strain changes can indicate the state of change of charge and health of batteries, helping to optimize performance and lifespan [22,23,24,25]. Material science leverages strain measurement to characterize and develop new materials, such as composites, smart materials, and nanomaterials, enabling advancements in high-strength, lightweight, and durable materials for various high-tech applications [26,27,28,29]. Strain sensors with low elastic modulus are being applied for tracking human body motions [30,31,32,33]. In geology, strain is measured at shallow depths to characterize deep geologic reservoirs [34].

Strain measurements can be categorized into contact and non-contact methods. For contact-based strain sensors, the traditional foil resistive strain gauge, invented in the 1930s, is the most widely used. It measures strain by detecting changes in electrical resistance of a metal foil bonded to a material’s surface. As the material deforms under stress, the foil gauge deforms with it, altering its electrical resistance proportionally to the strain. Strain gauges have extensive applications in material testing, structural component testing, and structural health monitoring [35]. However, this sensor can only measure strain at one position. The long installation times and electrical connections needed for data acquisition make it inefficient for multipoint measurements. Large-scale structures might need thousands of strain gauges for comprehensive measurements.

Piezoelectric sensors measure strain by converting mechanical deformation into electrical signals generated by piezoelectric materials [36], which generate a voltage in response to applied mechanical stress. These sensors are particularly effective for dynamic measurements and can be manufactured in various sizes, shapes, and complexity [37], making them suitable for monitoring structures with high vibrational frequencies, such as aircraft [38]. Applications have been reported that use a network of piezoelectric sensors for 2D strain mapping [39], but studies show that it is not advisable to use them for strain levels over 150 microstrains since their nonlinearities and changes in material properties with stress affect calibration accuracy [40]. Other sensor types, such as piezoresistive [41], capacitive [42,43], and triboelectric [44], are also used for strain sensing. These sensors have been used more for human body motion tracking to control robotics rather than for structural health monitoring and testing.

Fiber Bragg grating (FBG) sensors can be embedded inside structures and are widely used for structural health monitoring [5,45,46,47,48,49]. FBG sensors operate on the principle of measuring strain-induced changes in the wavelength of light reflected from a diffraction grating inscribed within an optical fiber [50]. When the fiber is subjected to strain, the grating’s periodicity changes, causing a shift in the reflected wavelength that is proportional to strain. Despite their benefits, FBG sensors are costly and require specialized equipment for installation and interrogation. They are sensitive to strain but can only measure along a specific direction and require careful alignment and calibration.

Another type of fiber optic sensor for strain measurement is based on Brillouin Optical Time Domain Reflectometry (BOTDR) [51] or Brillouin Optical Time Domain Analysis (BOTDA) [52]. These methods rely on the interaction between light and acoustic phonons in the fiber, known as Brillouin scattering. By analyzing the frequency shift of the backscattered light, BODTDR and BOTDA systems can measure strain and temperature along the entire length of an optical fiber, offering continuous strain measurement over long distances (up to several tens of kilometers) [53,54]. This makes the Brillouin-based methods ideal for long-range measurement on structures such as bridges, tunnels, and oil pipelines, but they are less suitable for 2D strain mapping.

Although contact-based measurement methods have been widely used for their simplicity and effectiveness in localized strain analysis, they fall short of providing a complete picture of structural behavior. This limitation underscores the importance of full-field strain mapping. The focus of this review is the importance and advancements of full-field strain mapping methods for experimental mechanics and structural health monitoring. Unlike single-point measurements, which provide localized data, full-field strain mapping offers comprehensive insights into the strain distribution across entire structures. This holistic approach is crucial for identifying stress concentrations, understanding material behavior under load, and detecting early signs of structural damage, thereby enhancing the accuracy and reliability of structural assessments. This paper will describe various strain mapping methods, especially optical techniques, including both traditional and emerging technologies. The discussion will range from the fundamental principles and measurement mechanisms to their practical applications and limitations of these methods in structural health monitoring and testing. Through this comprehensive review, the paper aims to highlight the critical role of full-field strain mapping in advancing structural analysis and ensuring the safety and longevity of structures.

## 2. Full-Field 2D Strain Mapping Methods

### 2.1. Digital Image Correlation (DIC)

DIC is a widely accepted and applied method for strain mapping. It captures digital images of an object under various load conditions. By identifying surface optical features in the images, DIC tracks the feature positions at different loading states by using correlation-based matching algorithms. The obtained feature displacements are then used to compute strain and ultimately generate the full-field strain responses of materials and structural components [55].

DIC techniques can be categorized into two main types, as illustrated in Figure 1. The two-dimensional (2D) DIC method employs a single camera to measure in-plane strain. This method places strict constraints on specimen out-of-plane motion, loading device alignment, and the relative positioning of the camera and the specimen in order to ensure accurate measurements [56]. It is inapplicable for non-planar surfaces or when 3D deformation occurs. To address these limitations, the three-dimensional (3D) DIC, or stereo-DIC, technique was developed. This uses two synchronized cameras or a single camera with specially designed light-splitting devices, based on the principle of binocular stereovision. Stereo-DIC accurately measures full-field 3D shapes and deformations of both planar and curved surfaces, making it more practical for real-world applications [57].

The process of determining surface strain using DIC involves four main steps. First, the test sample surface must show spatially dense intensity variations to serve as optical features for image registration techniques. If the sample does not have a suitable natural texture, artificial speckle patterns need to be applied to its surface [58,59]. Second, the cameras need to be calibrated to compensate for lens distortions and obtain the intrinsic and extrinsic parameters of the imaging system. Third, images of the test sample at different loading states are recorded by a single camera (for 2D-DIC) or two synchronized cameras (for stereo-DIC) and stored for post-processing. Lastly, for 2D-DIC, full-field displacements (in pixels) are obtained by comparing digital images of the planar sample at different states using subset-based local or Finite Element Method (FEM)-based global correlation algorithms [60,61]. For stereo-DIC, based on the calibrated parameters and disparity maps computed by DIC algorithms, 3D coordinates of measurement points are reconstructed. The subtraction of 3D points between two states provides the surface displacement fields, which can be smoothed and then differentiated or directly differentiated using numerical approaches to estimate strain distributions.

### 2.2. Carbon Nanotube (CNT)-Based Strain Sensors

CNTs, which can be further classified into single-wall carbon nanotubes (SWCNTs) and multiwall carbon nanotubes (MWCNTs), are of great interest for their unique fundamental properties and potential applications. The piezo-resistive properties of CNTs make them promising sensors for strain measurement [62,63]. Abot investigated yarns made from single-wall carbon nanotubes as piezo impedance-based strain sensors [64]. By embedding CNTs in polymers, flexible carbon nanotube strain-sensing films can be made [65]. Dharap tested the sensing performance of SWCNTs in polymer films under tensile, compressive, and flexural strain [66,67]. Li and Vemuru validated the feasibility of using MWCNT films for strain sensing [68,69]. Although these films can be used as sensor arrays to cover a large area, wires and connections are still needed for data acquisition and the resolution is limited.

Another way to use CNTs as strain sensors is based on their spectroscopic properties. The leading work in this area involves SWCNTs, which show systematic shifts in their vibrational and electronic spectra in response to mechanical deformation. SWCNTs that are attached to a surface can, therefore, be used as tiny, optically interrogated strain sensors. Several studies have demonstrated SWCNT-based strain sensing using shifts in the nanotubes’ vibrational Raman scattering frequencies [70,71,72,73]. However, although Raman scattering from SWCNTs is greatly enhanced by optical resonance with electronic transitions of the nanotubes, the scattering intensities are still low compared to most fluorescence emissions. This has hampered the speed and practical implementation of SWCNT Raman strain sensing.

Much stronger optical signals, faster data acquisition, and higher strain sensitivity have been obtained by using SWCNT near-IR fluorescence spectra to deduce strain. This is the basis of the “strain-sensing smart skin” or S^4^ method that has been proposed and demonstrated [74], as shown in Figure 2. In this method, emissions from SWCNTs embedded in a thin polymer film on the specimen surface are captured and spectrally analyzed to find the local strain magnitude at the desired locations and directions. Because the nanotube sensors are distributed across the entire coated surface, strain values can be measured at arbitrary locations and directions, and combined to give full-field strain maps [75,76,77,78].

In the S^4^ films, nanotubes are individually coated with the organic polymer PFO (poly(9,9-di-n-octylfluorenyl-2,7-diyl)) and applied by spraying or brushing onto the specimen surface as a toluene suspension. Solvent evaporation then leaves a submicron-thick sensing layer. After that, deformation of the specimen causes load transfer through the adhering polymer to the embedded SWCNTs, transmitting strain that is revealed by non-contact spectroscopic measurements of shifts in the SWCNT fluorescence peak wavelengths. The wavelength separation between the (7,6) and (7,5) peaks, illustrated in Figure 3, is found to be a reliable monitor of strain. Using a standard spectral gauge factor that relates these peak shifts to strains, one can measure the strain magnitude and the principal axis of strain at any location on the specimen surface by adjusting the fluorescence excitation beam position and polarization plane.

The sensing skin can have a “sandwich” structure containing optional top and bottom layers to isolate and protect the sensing layer from environmental damage. The multi-layer structure also lets the user paint a speckle pattern onto the bottom layer to allow parallel measurements of DIC and S^4^ on the same specimen. Figure 3b illustrates the three-layer coating structure.

Figure 3c shows the procedure used to measure strain maps using S^4^. First, before loading, an optional baseline map of the specimen can be obtained by point-wise raster scanning of the optical strain reader over the region of interest (ROI). At each point, the peak wavelengths of (7,6) (λ_(7,6)_) and (7,5) (λ_(7,5)_) SWCNT emissions are determined, and the wavelength difference is recorded as an element of a reference spectral array. After the specimen has been stress tested, it is scanned again with the same raster pattern to generate the final spectral array. The reference array elements are then subtracted from the final array elements and the result is divided by a pre-calibrated “spectral gauge factor, γ (−1.4 nm/mε) to obtain the array of induced strain values. Figure 4c shows an example of an S^4^ strain map of an aluminum bar with plastic strain.

The most recent S^4^ development uses a hyperspectral imaging system to make the measurements faster. As illustrated in Figure 5, the specimen coated with S^4^ film is illuminated with a red laser whose wavelength is chosen to efficiently excite the nanotube sensors. Their resulting near-infrared SWCNT emissions are imaged by a sensitive sCMOS camera through a high-aperture lens. Before entering the camera lens, the SWCNT emissions are spectrally selected by a specialized bandpass filter mounted on a computer-controlled rotation stage. Multiple images are captured at different filter angles that pass different spectral slices of the nanotube emission. A custom computer program then quickly analyzes the set of images to find the peak emission wavelength at each image pixel, convert the wavelength shift into strain using a known spectral gauge factor, and compile the results into a color-coded strain map of the specimen that can show more than 10^5^ independent measurements. This hyperspectral imaging S^4^ implementation reduces the measurement time per point by a factor of ca. 1000 compared to the sequential scanning method. More details and measurement results will be presented in a separate article.

### 2.3. Interferometric Methods

Interferometric methods utilize the principle of light interference to detect and quantify strains, making them good tools for experimental mechanics.

#### 2.3.1. Photoelasticity

Photoelasticity is a nondestructive, full-field optical technique used to measure stress in transparent objects subjected to loads. This method relies on the optical property known as “birefringence” or “double refraction”, which is present in many transparent polymers [80,81]. When a photoelastic sample/coating between two circular polarizers is loaded and illuminated with a conventional light source, it generates fringe patterns. These fringe patterns correspond to the differences between the principal stresses in a plane perpendicular to the direction of light propagation. This photoelasticity method is used mainly for visualizing stress distributions based on the fringe pattern [82,83]. Recent research enables non-destructive 3D photoelastic analysis by reconstructing the 3D stress tensor fields as neural implicit representations from polarization measurements, a technique known as NeST, as shown in Figure 6.

#### 2.3.2. Moiré Method

The Moiré method involves analyzing the interference patterns created from two superimposed periodic grids or gratings. When the object is placed under load, the spacing and orientation of the specimen grating change, resulting in a Moiré fringe pattern that visually reflects the strain distribution across the object’s surface [85,86]. Currently, the Moiré method for strain measurement primarily uses the scanning Moiré fringe (SMF) technique. In this approach, a reference grating is scanned across the specimen grating, which is attached to the deformed object [87], as shown in Figure 7. This scanning process produces detailed and continuous Moiré fringe patterns, allowing for precise measurement of strain and displacement. As an advanced variant of the traditional Moiré method, SMF is used for strain mapping on micro-scale structures, such as transistors, with nanometer-scale resolutions [88,89,90]. Other techniques such as the sampling Moiré methods [91] are also used for strain mapping on small-scale structures.

#### 2.3.3. Electronic Speckle Pattern Interferometry (ESPI)

When a coherent laser beam illuminates a material with an optically rough surface, it generates a speckle pattern characterized by random phase, amplitude, and intensity. As the material undergoes displacement or deformation, changes in the distance between the camera and the material surface alter the phase of the speckle pattern [92]. By analyzing these phase changes, ESPI can accurately quantify small deformations and strains across the specimen surface. This method has been used for studying the properties of metal fractures (Figure 8), microelectromechanical systems, and biological bones [92,93,94].

#### 2.3.4. Shearography

Shearography, also known as Electronic Speckle Shearing Interferometry, is a technique similar to ESPI but with a key difference in its approach. Instead of using a separate reference beam, shearography utilizes the surface of the test specimen itself as the reference. The interference pattern is generated by two sheared speckle fields that originate from the light scattered by the surface of the specimen under test [95]. Unlike holography and ESPI, which measure surface displacements, shearography measures the derivatives of surface displacements, which directly determine strain [96]. This technique is used for high-precision strain mapping and non-destructive testing [97,98,99,100], as shown in Figure 9.

#### 2.3.5. Digital Holography

Digital holography is a technique that captures and reconstructs the entire optical field, encompassing both intensity and phase [101]. It involves recording two optical wave fields at different times, which are then reconstructed simultaneously to create an interference pattern. This process, known as holographic interferometry, allows for the measurement of differences in optical phases caused by surface strain [102]. Digital holography is renowned for its ability to achieve nano-scale resolution and accuracy, making it particularly effective for strain measurement in electronic devices [25].

### 2.4. Other Methods

#### 2.4.1. Mechanoluminescence (ML)

ML refers to the emission of light when a mechanoluminescent material is subjected to stress. This phenomenon has numerous applications [103], including its use in optical sensing of stress [104]. The effective strain distribution on a structural surface can be inferred and visualized through the stress-induced photons emitted from an artificial skin composed of nano- and micro-ML particles embedded in a suitable polymer matrix. Various ML materials and methods are being explored for their potential in structural health monitoring, offering a novel approach to detect and analyze strain in different structures [105,106,107], as illustrated in Figure 10.

#### 2.4.2. Grid Method

The grid method is a strain mapping technique similar to DIC, but instead of using randomly distributed speckle patterns, the grid method uses regular patterns, as shown in Figure 11. A periodic grid is first transferred onto the specimen surface, and images of the grid are compared before and after deformation to calculate strain [108]. The grid method is used for strain mapping in experimental mechanics and has advantages over DIC for revealing sharp gradient changes [109].

#### 2.4.3. Diffractive Nanostructure Strain Mapping

Diffractive Nanostructure Strain Mapping is a novel strain mapping technique that was recently proposed for non-contact strain measurement [110]. This method leverages the diffractive properties of nanostructures, which change in response to applied strain. By applying a layer of periodic nanostructures to a structural surface and illuminating it with a specific light source, the resulting color changes can be observed and analyzed to determine strain, shown in Figure 12. Although still under development, this method shows promise for remote health monitoring of large-scale structures.

## 3. Performance Characteristics

### 3.1. Accuracy and Resolution

Accuracy and spatial resolution are often the top priorities when choosing among different measurement techniques. For some techniques, measurement accuracy is inversely correlated with measurement resolution. As an example, DIC is a subset-based approach based on local displacement smoothing with polynomials. Displacements are estimated at points, which are considered as the nodes of a mesh built up by the user. The displacement is then locally smoothed or interpolated by functions that are derived to provide the strain components [109]. This is usually not a concern when measuring large strains without sharp spatial gradients. However, it makes DIC poorly suited for studies and testing that need both high accuracy and high resolution, such as fracture mechanics on composite materials and damage detection on critical structures.

Figure 13 shows the strain distribution of a compressed concrete specimen with a hole at the center measured by S^4^ and 3D-DIC. DIC and S^4^ measurements reveal the resulting localized strain anomalies around the hole. However, in the DIC case, many strain features representing micro-cracks on the concrete are absent and masked by the noise of the measurement. Note that the loading level was controlled to give less than 2 × 10^−3^ strain (2 mε), critical for early-stage damage detection and structural health monitoring. A more detailed comparison has been reported [79]. The grid method also showed advantages for revealing strain distributions with steep gradients [109].

The accuracy of DIC results also depends on parameter tuning and the operator’s understanding of this technique. For instance, the virtual strain gauge (VSG) is a common and key element in the DIC method. VSG is a small region of the image over which average strain is calculated, analogous to the physical area covered by a resistive strain gauge. The choice of VSG size, therefore, determines the spatial resolution and accuracy in a DIC strain measurement. A small VSG size generates strain maps with less spatial smoothing at the cost of noisier strain data, while a large VSG size reduces noise but may fail to detect sharp spatial strain gradients indicative of structural damage. The choice of VSG often depends on the user’s estimation and judgment. Figure 14 shows the strain distribution in an acrylic bar with a square notch stressed in tension, as measured by S^4^ and DIC and simulated by FEM. The localized strain maxima indicated by the two red spots appear diffuse and displaced in the DIC strain map. Figure 15 shows the influence of different VSG values on the resolution of the strain map.

There are other similar parameters in DIC. In practical applications, a sensitivity study with a reference measurement using a strain gauge is usually required to determine those parameters. However, such references can be challenging to obtain in many applications. By contrast, the errors induced by subjective factors and post-processing are minimal for S^4^. As a direct method, S^4^ deduces strain from spectral shifts using a pre-determined gauge factor that does not need calibration before measurement. Additionally, each scanned point or image pixel measurement is independent of the others, giving a normal spatial resolution of 0.1 mm for the scanning-based system and 0.2 mm for the camera-based system.

Interferometric methods are effective for strain measurements on small and micro-scale structures. Digital holography can attain nano-scale resolution with an approximate strain error of less than 10 × 10^−6^ strain (10 με) [111], making it ideal for applications requiring detailed strain measurements in micro-scale structures and electronic devices. The major error source in this case is imprecise alignment and setup of optical components. The minimum measurable strain of ESPI relates to the field of view, with larger fields of view allowing greater measurement sensitivity [93]. For ideal applications, the strain error margin is typically considered to be around 50 × 10^−6^ (50 με) [112], but its accuracy can be significantly influenced by environmental vibrations and noise. Shearography, while less sensitive to environmental disturbances, offers robust measurement capabilities for larger structures with an uncertainty of about 100 × 10^−6^ (100 με) [113] and provides direct strain information by measuring the derivatives of surface displacements. The primary error source for shearography is the accurate interpretation of shearograms, which requires proper handling and alignment. The Moiré method, including its advanced scanning variation, delivers strain and displacement measurements with errors typically around 1 × 10^−3^ strain (1 mε) [85]. The accuracy of Moiré techniques can be affected by the quality of the gratings and the stability of the setup. Overall, the choice of interferometric method depends on the specific application requirements, including the desired resolution, sensitivity, and environmental conditions.

The ML method is typically used for stress mapping on different sizes of structures [104]. There is still limited research that can accurately quantify strain based on ML light intensities. Azad et. al. calibrated a thin-film ML sensor and developed a full-field quantitative strain measurement with pixel-level resolution. The ML sensing film correctly captured the strain concentrations, but the authors also noted that the film is not sensitive enough to react to small strains (about 100 με) and low strain rates (about 340 με·s^−1^) [107]. Shin et al. combined a DIC algorithm with ML to quantify the strain [105]. The ML field can be linearly mapped to a strain field by a pre-calibrated gauge factor. A comparison between ML and DIC strain maps around a crack tip at sub-millimeter resolution is shown in Figure 16.

### 3.2. Dynamic Measurement

Methods using digital cameras as sensors, such as DIC and the interferometric methods, are naturally suited for dynamic strain mapping due to their ability to capture changes in deformation at frequencies from a few Hz to hundreds of Hz. However, there is an inherent trade-off between resolution and frame rate: higher resolution typically requires lower frame rates and reduced bandwidth for dynamic measurements. The solution usually involves using high-quality cameras supported by computers with powerful hardware. This substantially increases the cost of systems, especially those using multiple cameras, such as for 3D-DIC. Moreover, synchronization of two high-speed cameras can be challenging and even unachievable for some ultra-high-speed cameras. Solutions have been proposed and developed to push the measurable frequency limit into the kHz range [100,114,115,116].

Although data acquisition in the S^4^ method is intrinsically slower than in DIC and interferometric methods, S^4^ can capture oscillatory strains at frequencies of 5 Hz [117] using the scanning configuration held at a single point on a specimen. Hardware modifications may allow single-point S^4^ measurements at higher frequencies. The hyperspectral imaging version of S^4^ requires a few minutes of data acquisition and processing to generate one strain map. Planned refinements are expected to improve speed significantly and permit some dynamic measurements.

### 3.3. Long-Term Structural Health Monitoring

Strain mapping requirements differ between experimental mechanics and structural health monitoring due to scale, objectives, and operational constraints. Experimental mechanics focuses on high-precision strain measurement with high spatial resolution to capture detailed local material behavior in controlled environments. In contrast, structural health monitoring prioritizes long-term, global, and local continuous monitoring of large structures in locations with limited access. The sensors must be durable under relevant environmental conditions and should not affect the structure’s appearance and function.

For effective long-term global-local structural health monitoring, strain mapping methods need several key features. First, high accuracy and resolution are critical to detect local strain concentrations that accumulate over time and may cause significant damage. This ensures that even minor defects or early signs of damage are identified promptly for proper maintenance. Second, robustness and durability are important, as the system must withstand varying environmental challenges, including sunlight, temperature variations, humidity, pollutants, corrosion, and potential mechanical disturbances. Third, surface disruption should be avoided to protect the structure’s integrity without interrupting normal operations. Finally, long-term reliability and stability are necessary to ensure consistent performance with minimal maintenance over the monitoring period. Incorporating these features will enhance the effectiveness of strain mapping techniques in ensuring the safety and longevity of critical structures.

**Accuracy and resolution**: DIC is effective for measuring large strains but generally lacks the sensitivity needed to detect small localized strains and high gradients in strain maps, which are crucial for early damage detection and evaluation. Additionally, the results from DIC can be operator-dependent, with extensive training needed to obtain reliable strain maps. The ML and interferometric methods offer high resolution and accuracy, but their performance relies heavily on advanced data post-processing algorithms. Data processing in the S^4^ method is highly automated and does not depend on operator judgment. S^4^ can detect small localized strain values, and because measurements at different points are independent, it can properly map sharp strain gradients near edges or cracks.

**Robustness and stability**: Interferometric methods require geometrically stable setups in controlled environments to maintain accuracy, making them less adaptable to harsh or fluctuating conditions. In contrast, DIC speckles are easy to make and exhibit strong environmental adaptability, functioning effectively in challenging environments such as high pressure and temperature [118,119]. The environmental robustness of other methods, such as S^4^ and ML, still requires further study. The application of protective top coats over the sensing films offers a potential solution for enhancing durability and reliability in adverse conditions.

**Surface disruption**: DIC can negatively affect the surface appearance of structures through the application of black-and-white speckle patterns. Such coatings may not be acceptable for uses in which visual aesthetics are important. Some materials used in the ML coatings have their own inherent color, which might affect the appearance of the structure. S^4^ coatings are clear and virtually colorless, providing a suitable option for strain sensing of structures whose appearance must be undisturbed.

**Long-term reliability**: Interferometric methods are generally not well suited for extended monitoring due to their sensitivity to environmental changes, which can affect measurement accuracy over time. DIC also poses challenges for long-term measurement applications because it is a reference-based method that requires stable or repeatable positioning of the camera system and precise image registration relative to the object. The S^4^ method offers a significant advantage because its strain measurements do not rely on precise image registration. This means that the method can easily be used for periodic strain mapping to detect damage induced during service in mobile objects such as aircraft or other structures in dynamic motion.

Even though there are many methods available for full-field 2D strain mapping in experimental mechanics, only a few are suitable for or have been applied in structural health monitoring. Because there are various structure types, each serving a different purpose, no single method can meet all structural health monitoring requirements. A system that integrates multiple types of sensors will likely provide the optimal solution. For example, combining optical fiber sensors with S^4^ could enhance the monitoring of bridges. Optical fiber sensors can be applied over long distances and used for real-time monitoring of the overall global behavior of the structure. When abnormal behavior is identified, S^4^ can help locate and assess detailed localized damage sites or defects, facilitating targeted maintenance efforts. Multiple probes and cameras can be used together for large-area measurements.

As a summary, a comparison of different methods is shown in Table 1.

## 4. Challenges

Although strain mapping techniques are evolving steadily, there is still a gap between research and real-world applications. Several future challenges must be addressed to enhance their reliability and applicability.

**DIC**: One challenge is performing DIC strain measurements by imaging the natural surface patterns of materials with weak optical features rather than using applied speckle paint patterns. Additionally, when processing DIC images, it is important to identify the optimal subset size and shape function for subset matching to improve accuracy and reduce errors in the correlation process. Adaptive algorithms that can adjust these parameters in real-time are needed. Similarly, adaptive selection of the optimal strain window size and shape function is desired to accurately measure strain, especially in regions with high strain gradients [120,121].

**Interferometric Methods**: One of the primary challenges is simplifying the complex measurement setups that are required. Reducing the number of components and alignment requirements will make these techniques more accessible and user-friendly. Another concern is that interferometric methods are highly sensitive to environmental factors such as temperature fluctuations and vibrations. Developing methods to minimize these sensitivities will enhance the reliability of these measurements in various conditions.

**ML Methods**: A major challenge is developing reliable techniques for converting ML intensity maps into quantitative strain maps. This will require research leading to effective algorithms to accurately perform this correlation, which will be needed to expand the applicability of ML methods.

**S^4^ Method**: Improving the measurement speed and enlarging the mapped area seem to be the main goals for more practical S^4^ strain mapping. It is also necessary to demonstrate the durability and stability of S^4^ coatings under realistic environmental conditions.

Addressing these challenges will significantly enhance the capabilities of each strain mapping technique, making them more versatile and reliable for various applications in experimental mechanics and structural health monitoring.

## 5. Conclusions

In summary, non-contact strain mapping is a critical tool in experimental mechanics and structural health monitoring, providing essential insights into structural behavior and integrity. Each of the available strain mapping techniques offers unique advantages and disadvantages, necessitating careful selection based on specific application needs. Interferometric methods are particularly suited for small-scale measurements within controlled laboratory environments due to their high precision but are less adaptable to industrial scenarios. Digital image correlation is widely used due to its ease of application and flexibility; however, its results can be operator-dependent, and it is not ideal for long-term monitoring of deployed structures. Mechanoluminescence methods, which do not require an artificial surface pattern, provide high sensitivity; however, converting signals into quantitative strain maps remains a significant challenge. The strain-sensing smart skin (S^4^) method shows advantages in accuracy and spatial resolution for detecting localized sharp gradient changes; its results do not vary with operator expertise, and it is well suited for long-term structural health monitoring despite its current limitations in dynamic strain map measurements. Therefore, selecting the appropriate strain mapping method depends on identifying those whose capabilities best match the particular application requirements.

## Figures and Tables

**Figure 1 sensors-24-06573-f001:**
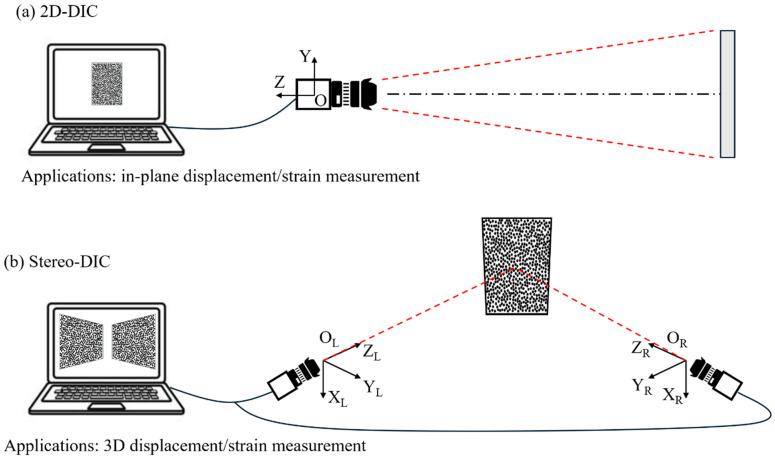
Two typical DIC techniques: their imaging systems and applicability.

**Figure 2 sensors-24-06573-f002:**
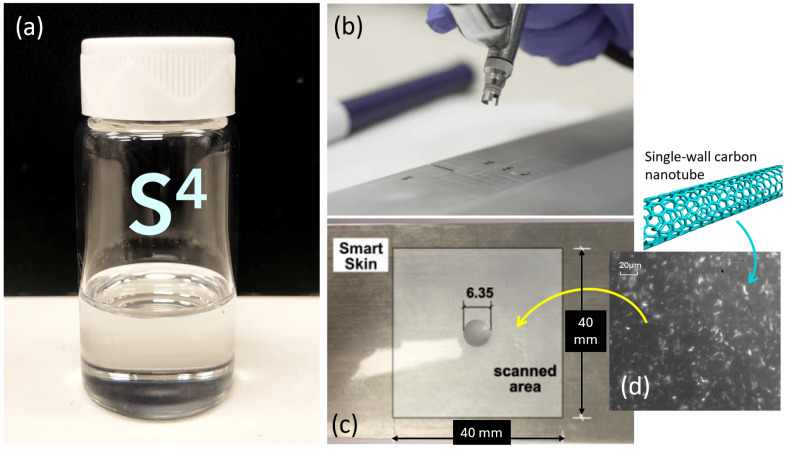
(**a**) S^4^ solution containing suspended SWCNTs in toluene an PFO; (**b**) Coating S^4^ on an aluminum specimen by airbrushing; (**c**) S^4^ film coated on an aluminum specimen around the hole; (**d**) fluorescence micrograph of the S^4^ film with randomly and uniformly dispersed SWCNTs.

**Figure 3 sensors-24-06573-f003:**
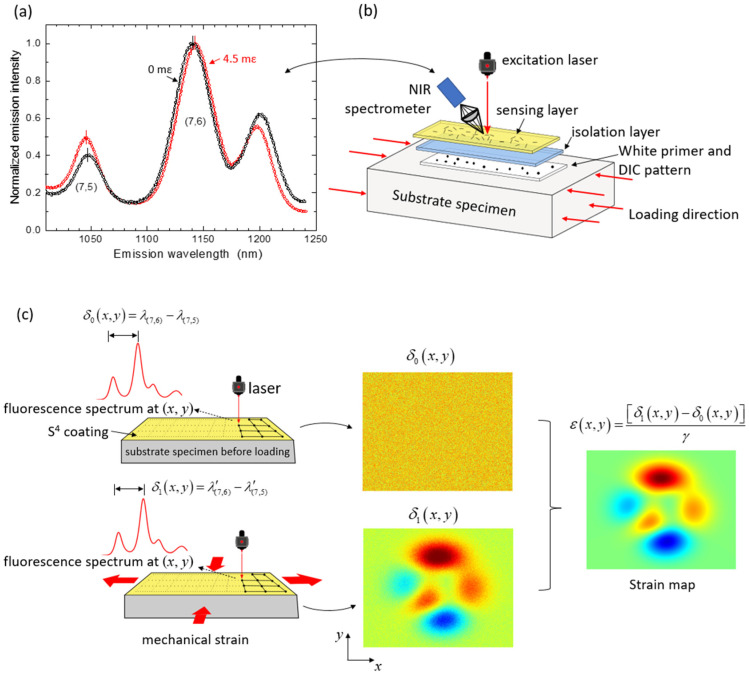
(**a**) Emission spectrum from an S^4^ film on a specimen with substrate strain of 0 (black points) and 4.5 mε (red points); (**b**) layer-structured S^4^ film; (**c**) scheme for measuring 2D strain maps in the S^4^ method. The specimen is raster-scanned before (left top) and after (left bottom) stress testing to find the spectral peak separation at each grid point. Those separations are then pointwise subtracted and divided by the spectral gauge factor to obtain the specimen’s strain map (right) [79].

**Figure 4 sensors-24-06573-f004:**
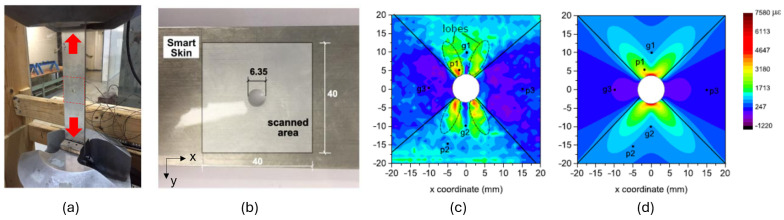
(**a**) An aluminum bar with a hole at the center under tensile stress; (**b**) S^4^ coating is applied around the hole before loading; (**c**) residual strain distribution measured by S^4^; (**d**) residual strain distribution calculated by FEM simulation.

**Figure 5 sensors-24-06573-f005:**
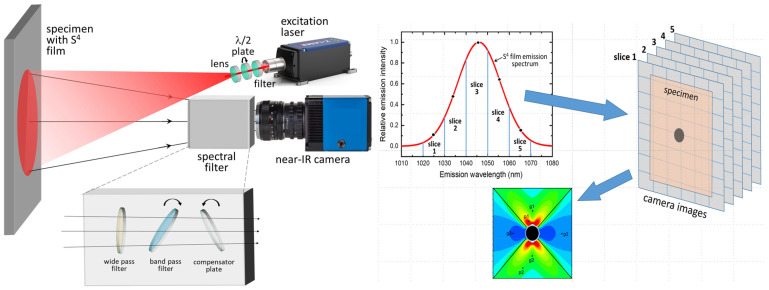
Hyperspectral imaging system for more efficient S^4^ strain mapping.

**Figure 6 sensors-24-06573-f006:**
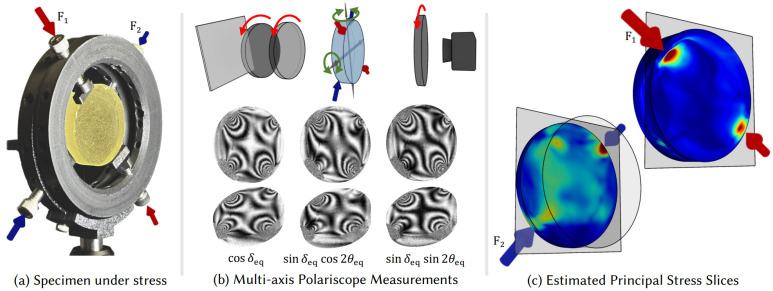
Overview of NeST (neural stress tensor tomography). (**a**) a transparent object subject to external force; (**b**) the object exhibits birefringence that manifests as fringes under multi-axis polariscope measurements; (**c**) the estimated principal stress slices reconstructed by the NeST method [84].

**Figure 7 sensors-24-06573-f007:**
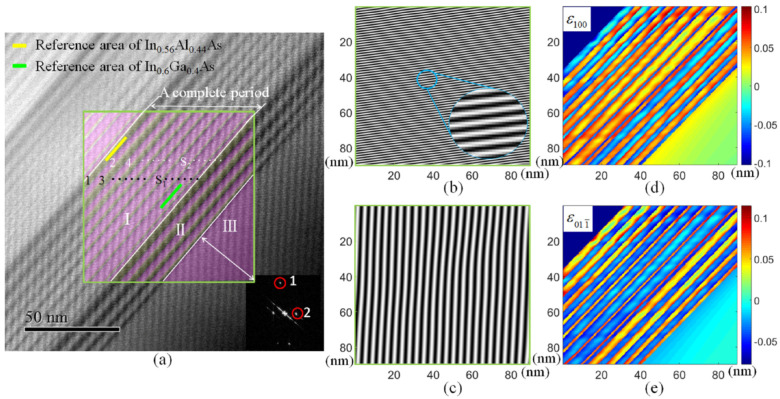
Strain mapping by the scanning Moiré fringe (SMF) process: (**a**) two-directional SMF imaging; (**b**,**c**) the unidirectional Moiré fringe decoupled from the 2D Moiré fringes in the green rectangle in (**a**); (**d**,**e**) the strain distributions obtained in different directions [90].

**Figure 8 sensors-24-06573-f008:**
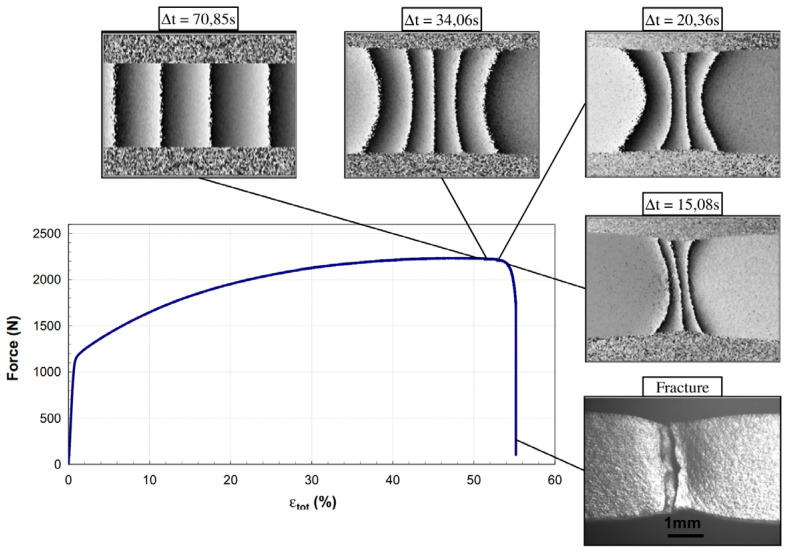
The evolution of fringe patterns from electronic speckle pattern interferometry (ESPI) during tensile testing of a stainless steel specimen and the photo of its fracture [94]. The Δt indicated the lapse of time between the two-phase maps used to compute the pattern.

**Figure 9 sensors-24-06573-f009:**
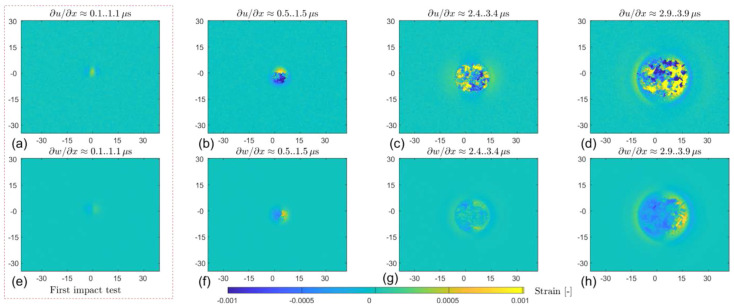
In-plane (*∂u/∂x*) and out-of-plane (*∂w/∂x*) surface strain maps of a 4 mm aluminum plates (6082-T6) at times from 0 to 2.9..3.9 μs (**a**–**h**) measured by shearography during an impact test [100].

**Figure 10 sensors-24-06573-f010:**
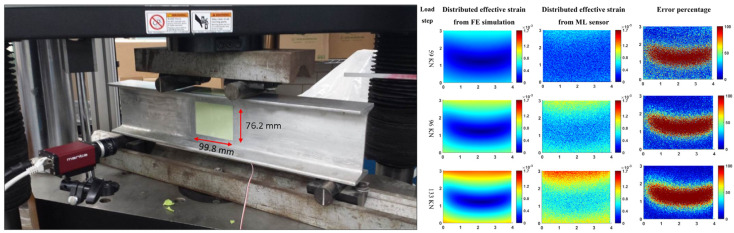
(**Left**) Four-point bending test on aluminum I-beam specimen coated with a mechanoluminescent (ML) sensing coating. (**Right**) simulated and measured strain distributions from FEM and ML [107].

**Figure 11 sensors-24-06573-f011:**
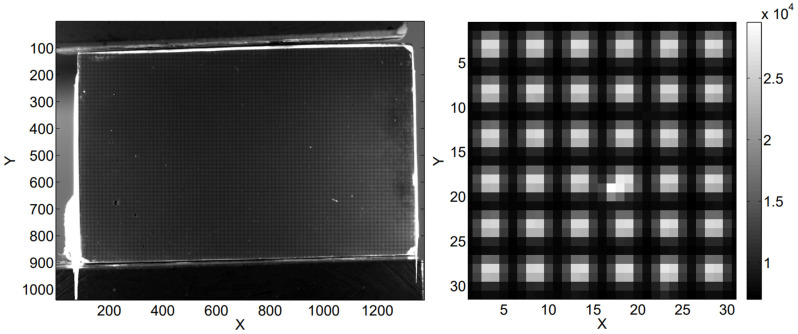
(**Left**) A specimen prepared for grid method strain mapping. (**Right**) locally magnified grid.

**Figure 12 sensors-24-06573-f012:**
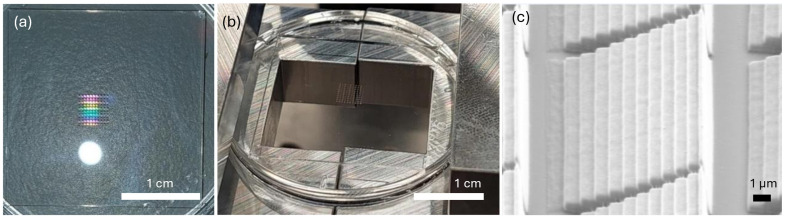
Diffractive nanostructure strain mapping. (**a**) The fabricated nanostructure on a glass substrate; (**b**) a mold on top of an aluminum sample holder; (**c**) a unit cell imaged with an electron microscope.

**Figure 13 sensors-24-06573-f013:**
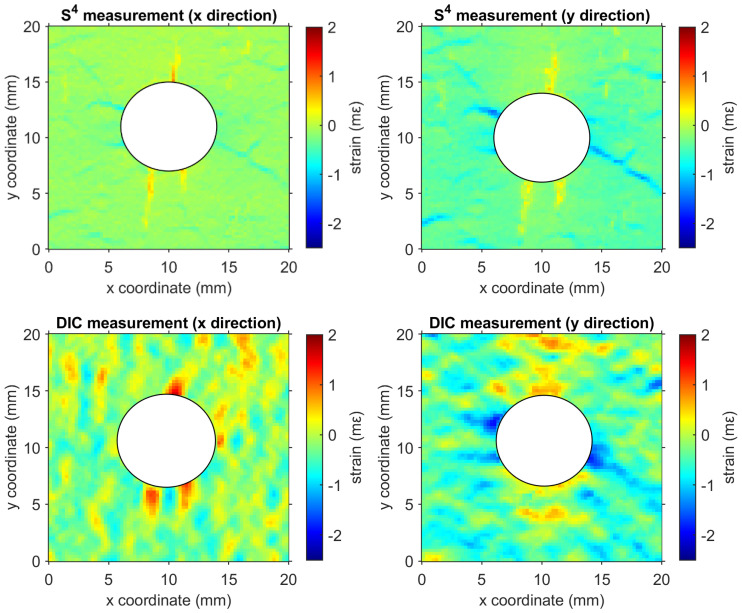
Strain maps of a concrete specimen with a central hole that was compressed along the y-axis. Rows from top to bottom show strain measured with S^4^ and 3D-DIC. The left panels show strain components perpendicular to the stress axis, and the right panels show components parallel to the stress axis [79].

**Figure 14 sensors-24-06573-f014:**
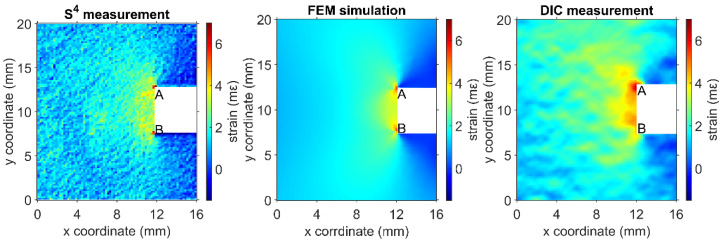
Strain maps of a notched acrylic specimen under tensile stress measured with S^4^ and DIC, and the corresponding FEM simulations.

**Figure 15 sensors-24-06573-f015:**
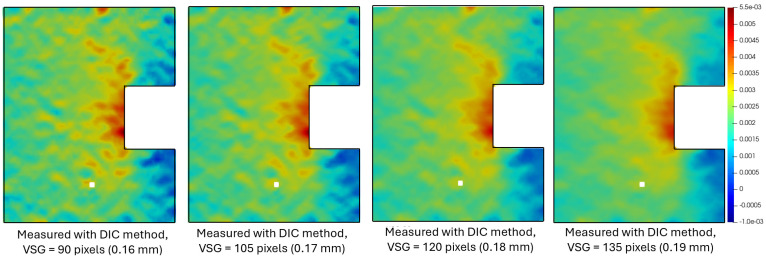
DIC strain maps of a notched acrylic specimen under tensile stress deduced using the same image data with different virtual strain gauges (VSGs).

**Figure 16 sensors-24-06573-f016:**
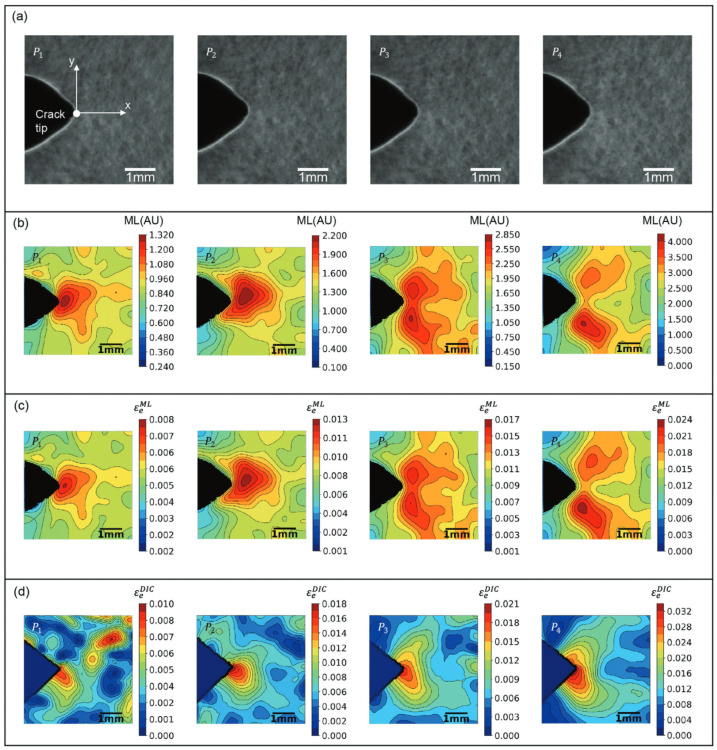
Visualizations from mechanoluminescence (ML) and digital image correlation (DIC) of the effective strain field near a crack tip. (**a**) Sequential images of the specimen under different loads; (**b**) Sequence of ML images; (**c**) Sequence of strain contour maps deduced from the ML intensity maps; (**d**) Sequence of DIC strain maps [105].

**Table 1 sensors-24-06573-t001:** Performance comparison of the major strain mapping methods.

	DIC	S^4^	Interferometry	Mechanoluminescence
**Strain accuracy**	Good for large strains > 10 mε, less accurate for small strain measurements. Accuracy is operator-dependent.	Good for strains from 100 με to 10 mε. Accuracy not dependent on operator skill.	Good for small strains. Accuracy varies from 10 με to 1 mε, depending on method.	Good for strains > 1 mε since the ML material is not sensitive to small strains (100 με). Accuracy calibration is challenging.
**Spatial resolution**	Wide range, but not good for revealing localized sharp spatial gradients. Spatial resolution can be traded for strain accuracy.	Camera-based system: typically 0.2 mm. Scanning system: wide range, down to less than 0.1 mm.	Very fine resolution, can be used at μm and even nm level.	Current applications are at mm or cm level of resolution.
**Measurement speed**	Good for dynamic measurements, with frequency range from a few Hz to kHz, depending on camera. However, camera synchronization is challenging for high-speed measurements.	More suited for static or quasi-static measurements with ~1 or 2 min needed per strain map, but ~0.1 s for single points. Pulsed lasers may allow strobed measurements.	ESPI and Digital Holography suitable for dynamic measurements; Shearography and classical interferometry typically used for static or quasi-static measurements.	More suited for static and quasi-static measurements. Dynamic response time and accuracy depend on the ML material sensitivity and calibrations.
**Post-processing**	Slow, with need for manual parameter tuning requiring operator’s judgment.	Automatic and fast, in a small fraction of the time needed for data acquisition.	Depends on method: ESPI is fast but Digital Holography and classic interferometry are slow.	Strain inferred from stress values that are found quickly from emission intensity and calibration.
**Size of measured objects**	Flexible, from small to large structures; limited by camera and speckle size.	From small to large structures with multiple hyperspectral cameras; limited by measurement time.	Optimal for small structures (e.g., microelectronics).	Flexible, from small to large structures.
**Sensor environmental durability**	Very durable, even at high temperatures and humidity.	Durable for normal outdoor and indoor conditions.	Low durability, suitable for laboratory environments.	Durable for normal outdoor and indoor conditions.
**Surface alteration**	Significant effect on appearance from the applied speckle pattern.	Minimal; sensing film is thin and transparent.	ESPI, Digital Holography, and Shearography have no effect. Grid-based methods, such as sampling Moiré, alter surface appearance.	Small effect from the faint colors of typical ML materials.
**Long-term monitoring**	Less suitable. As a reference-based method, it needs careful realignments to keep accurate image registration.	Suitable. A reference-free method, so does not need continuous observation or precise image registration.	Not suitable. The system is very sensitive to vibrations and environmental factors.	Suitable. Does not need continuous observation or precise image registration.

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
