# Peer review of "A Review: Non-Contact and Full-Field Strain Mapping Methods for Experimental Mechanics and Structural Health Monitoring"

_sensors, 2024, doi:10.3390/s24206573_

Round 1
Reviewer 1 Report
Comments and Suggestions for Authors
This manuscript offers a thorough review of strain mapping methods, covering various techniques such as DIC, interferometric methods, and the grid method. It is suggested to be accepted with minor revisions. Specific suggestions are outlined below:
- The organization of the review needs to be improved. For instance, while section 2 is titled "Strain Mapping Methods," sections 3 and 4 also cover mapping techniques, which creates some overlap.
- The manuscript primarily focuses on non-contact strain measurement techniques. It is recommended to clarify this emphasis in the abstract.
- The sections on "Performance Characteristics" and "Challenges" do not comprehensively address all methods discussed. A comparison of the performance of all mentioned techniques and a discussion on the future development of strain mapping methods would strengthen the manuscript.
- A thorough proofreading is needed to correct several typographical errors. For example, the title of section 2 should be "Strain Mapping Methods."
- A thorough proofreading is needed to correct several typographical errors. For example, the title of section 2 should be "Strain Mapping Methods."
Author Response
Comments 1: The organization of the review needs to be improved. For instance, while section 2 is titled "Strain Mapping Methods," sections 3 and 4 also cover mapping techniques, which creates some overlap.
Response 1: Thank you for your insightful comment. We agree that the organization could be improved. To address this, we have restructured the paper by incorporating the original Section 3 (Interferometric Methods) and Section 4 (Other Methods) under Section 2 (Full-field Strain Mapping Methods), as they both cover strain mapping techniques. Additionally, we have added section numbers before each title to enhance clarity and make the structure of the paper clearer. These changes can be found at the beginning of each section title.
Comments 2: The manuscript primarily focuses on non-contact strain measurement techniques. It is recommended to clarify this emphasis in the abstract.
Response 2: We agree that the focus on non-contact strain mapping techniques should be clarified. To address this, we have added the term “non-contact” in both the title and abstract to clearly define the scope of the manuscript. These changes are reflected in line 1 and line 14 on page 1.
Comments 3: The sections on "Performance Characteristics" and "Challenges" do not comprehensively address all methods discussed. A comparison of the performance of all mentioned techniques and a discussion on the future development of strain mapping methods would strengthen the manuscript.
Response 3: Thank you for your insightful comment. We agree that a more comprehensive comparison of performance characteristics across all methods would strengthen the manuscript. To address this, we have included a performance comparison table of the major strain mapping methods, offering a clear and direct comparison of their advantages and limitations. This will help readers select the most suitable method for their specific applications. The new table and related discussion can be found at line 458 on page 19.
Comments 4: A thorough proofreading is needed to correct several typographical errors. For example, the title of section 2 should be "Strain Mapping Methods."
Response 4: Thank you for highlighting this issue. We have corrected the typographical error in the title of Section 2 at line 104 on page 3. Additionally, we have conducted thorough proofreading of the entire manuscript to ensure that all typographical errors have been addressed.
Reviewer 2 Report
Comments and Suggestions for Authors
1. Compared to the description of fiber Bragg grating sensing technology given in the introduction, distributed fiber optic strain sensing testing technology is a more advanced and rapidly developing type of fiber optic strain testing technology, such as Brillouin optical time domain reflection/analysis (BOTDR/A) technology, optical frequency domain reflection (OFDR)technology, etc., which can achieve one-dimensional scale strain mapping. It is strongly recommended to supplement relevant content on distributed fiber optic strain sensing testing technology
2. The 'stain' in the title of the second section should be 'strain'. Since the article title highlights the full field strain mapping method, it is suggested to add “full field” as an emphasis here. In addition, the article should define the full field strain mapping method, that is, whether it is different from the general strain mapping method.
3. From the perspective of the entire text, the description of the paper mainly focuses on aspect full-field strain mapping methods for experimental mechanics. However, it lacks a relevant comprehensive literature review on structural health monitoring. It is suggested to provide relevant descriptions for structural health monitoring recommendations, especially highlighting their main differences from experimental mechanics. Taking structural health monitoring in civil engineering as an example, such as bridges, tunnels, etc., whose engineering scale is generally relatively large and differs from refined experimental mechanics testing,
4. Suggest adjusting and beautifying the scale of the middle image and text in Figure 2, and indicate the units in Figure 2. The relevant symbols appearing in Figures 3, 7, etc. should be described in the manuscript. The symbol Δ t in Figure 8 should be explained in the text.
Author Response
Comments 1: Compared to the description of fiber Bragg grating sensing technology given in the introduction, distributed fiber optic strain sensing testing technology is a more advanced and rapidly developing type of fiber optic strain testing technology, such as Brillouin optical time domain reflection/analysis (BOTDR/A) technology, optical frequency domain reflection (OFDR)technology, etc., which can achieve one-dimensional scale strain mapping. It is strongly recommended to supplement relevant content on distributed fiber optic strain sensing testing technology.
Response 1: Thank you for your valuable suggestion. We agree that distributed fiber optic strain sensing, such as BOTDR/A and OFDR technologies, plays a significant role in structural health monitoring. In the revised manuscript, we have added a paragraph after the Fiber Bragg Grating sensor section to explain the basic principles of BOTDR/A and their applications in structural health monitoring. However, as the focus of this manuscript is on 2D strain mapping, and BOTDR/A sensors are primarily suited for long-distance 1D measurement, so they are not included in the subsequent discussions. These changes can be found from lines 82 to 89 on page 3.
Comments 2: The 'stain' in the title of the second section should be 'strain'. Since the article title highlights the full field strain mapping method, it is suggested to add “full field” as an emphasis here. In addition, the article should define the full field strain mapping method, that is, whether it is different from the general strain mapping method.
Response 2: Thank you for this helpful comment. We have corrected the typo in line 104 on page 3. We also agree that emphasizing "full-field" and defining it early on would help differentiate it from general strain measurement methods. Accordingly, we have added a definition of non-contact, full-field strain mapping in the abstract to clarify its distinction. These changes can be found at line 14 on page 1.
Comments 3: From the perspective of the entire text, the description of the paper mainly focuses on aspect full-field strain mapping methods for experimental mechanics. However, it lacks a relevant comprehensive literature review on structural health monitoring. It is suggested to provide relevant descriptions for structural health monitoring recommendations, especially highlighting their main differences from experimental mechanics. Taking structural health monitoring in civil engineering as an example, such as bridges, tunnels, etc., whose engineering scale is generally relatively large and differs from refined experimental mechanics testing.
Response 3: We appreciate this suggestion. Many of the 2D strain mapping methods discussed are still in early research stages and have yet to be widely implemented in real-world structural health monitoring applications. Consequently, there is limited literature on the use of 2D strain mapping for structural health monitoring in most of the methods covered. However, we agree that structural health monitoring, especially in civil engineering (e.g., bridges, tunnels), presents different requirements compared to experimental mechanics, particularly due to the larger scale and environmental factors. To address this, we have added a paragraph explaining the key differences in requirements between experimental mechanics and structural health monitoring. Additionally, we propose combining different methods to address practical challenges in large-scale applications. These changes can be seen from lines 402 to 408 on page 17 and lines 447 to 456 on page 18.
Furthermore, to strengthen the manuscript, we have added a performance comparison table of the major strain mapping methods. This table provides a clear comparison of the methods' pros and cons, allowing readers to select the most appropriate method for their specific applications. The new table can be found at line 458 on page 19.
Comments 4: Suggest adjusting and beautifying the scale of the middle image and text in Figure 2, and indicate the units in Figure 2. The relevant symbols appearing in Figures 3, 7, etc. should be described in the manuscript. The symbol Δ t in Figure 8 should be explained in the text.
Response 4: We have adjusted the scale and improved the layout of Figure 2, including adding the necessary units. Additionally, the relevant symbols in Figures 3, 7, and 8 have been further explained in the manuscript. The changes to Figure 2 can be found on page 5. In Figure 3, λ(7,5) and λ(7,6) represent the wavelengths of the peak positions for (7,5) and (7,6), respectively, and γ refers to the spectral gauge factor; these updates can be found in lines 181 and 185 on page 6. In Figure 8, Δt indicates the lapse of time between the two phases, as explained in line 254 on page 10.
Reviewer 3 Report
Comments and Suggestions for Authors
The presented review describes full-field strain mapping methods and a variety of strain sensors including Digital Image Correlation, Interferometric Methods, Mechanoluminescence, CNT- based strain sensors, diffractive nanostructure Strain Mapping etc. and discusses the Performance Characteristics obtained using the mentioned above methods. A special attention is paid to a long-term structural health monitoring and to challenges arisin for each strain mapping technique.
To my opinion, the review is well writted and organised and will be interesting to readers dealing with solid state physics and health science.
I believe that the paper can be publish with a minor correction.
I disagree only with one sentence in the review on Page 4 "Unfortunately, such Raman methods are hampered by intrinsically weak scattering signals and long measurement times, making them impractical for most applications." Raman signal from CNTs is very intense in comparison with a variety of materials. Therefore, I would propose to find another arguments which hamper a wide application of Raman spectroscopy for the strain analysis.
Author Response
Comments : I disagree only with one sentence in the review on Page 4 "Unfortunately, such Raman methods are hampered by intrinsically weak scattering signals and long measurement times, making them impractical for most applications." Raman signal from CNTs is very intense in comparison with a variety of materials. Therefore, I would propose to find another arguments which hamper a wide application of Raman spectroscopy for the strain analysis.
Response : We agree that resonance enhancement of Raman scattering from SWCNTs makes signals much more intense than Raman from most other materials. Nevertheless, that emission is still much weaker than fluorescence, even from low quantum yield samples including SWCNTs. In our view, this has made commercial implementation of Raman-based SWCNT strain measurement impractical. We have edited the text on page 4 (line 150-152) to clarify these points.
Reviewer 4 Report
Comments and Suggestions for Authors
First of all, the abstract of this paper shown on the website and in the manuscript is different. This should not have happened.
This manuscript aims to provide a comprehensive comparison of strain mapping methods used in experimental mechanics and structural health monitoring. However, this work fell short in the following aspects:
1. The contribution of this review work should be clarified. Inline 11, the authors stated "Despite the development of numerous strain mapping methods, a comprehensive comparison of these techniques is not found."
There have been books focusing on optical metrology that cover most of the methods considered in this manuscript, such as this one: https://iopscience.iop.org/book/edit/978-0-7503-3027-5
The Carbon Nanotube (CNT) based Strain Sensors, which was given much attention by the authors, have also been covered in review papers, such as this one: https://onlinelibrary.wiley.com/doi/full/10.1155/2012/652438
2. The aspect that might be unique to this review work is the focus on applications in experimental mechanics and structural health monitoring (SHM). However, the related discussions in sections 5 and 6 are rather general and qualitative, without specific consideration of concrete applications, particularly for SHM applications. One might ask, what if the visual/optical/physical access to the structure is limited which is usually the case for SHM application scenarios?
3. This paper gives an overview of strain mapping methods but does not provide a comprehensive comparison. Several examples were given comparing S4 and ML with FE simulation and DIC results. What about the other methods? As a minimum, a table should be added to summarise and compare the mentioned methods.
Author Response
Comments 1: The contribution of this review work should be clarified. Inline 11, the authors stated "Despite the development of numerous strain mapping methods, a comprehensive comparison of these techniques is not found." There have been books focusing on optical metrology that cover most of the methods considered in this manuscript, such as this one: https://iopscience.iop.org/book/edit/978-0-7503-3027-5 .
The Carbon Nanotube (CNT) based Strain Sensors, which was given much attention by the authors, have also been covered in review papers, such as this one: https://onlinelibrary.wiley.com/doi/full/10.1155/2012/652438
Response 1: Thank you for your insightful comment. We agree that there are existing works reviewing various strain mapping methods, but most focus on a specific category. For instance, optical metrology books primarily discuss interferometric methods, while CNT-based sensors are the focus of CNT review papers. However, we have not found a paper that compares methods across different categories, such as CNT sensors, interferometric techniques, ML and DIC. A comparison of methods from different categories could provide more valuable insights for users seeking the most appropriate technique for their specific strain mapping applications. To clarify this, we have revised the statement “Despite the development of numerous strain mapping methods, a comprehensive comparison of these techniques is not found” to “Although reviews have focused on specific methods, such as interferometric techniques or carbon nanotube-based strain sensors, a comprehensive comparison that evaluates these diverse methods together is lacking.” The changes can be seen in the Abstract on page 1.
Comments 2: The aspect that might be unique to this review work is the focus on applications in experimental mechanics and structural health monitoring (SHM). However, the related discussions in sections 5 and 6 are rather general and qualitative, without specific consideration of concrete applications, particularly for SHM applications. One might ask, what if the visual/optical/physical access to the structure is limited which is usually the case for SHM application scenarios?
Response 2: Thank you for your thoughtful feedback. In the manuscript, we have provided detailed comparisons of strain mapping applications in experimental mechanics, such as between carbon nanotube sensors and DIC, as well as DIC and ML, analyzing their performance differences and the underlying reasons. However, strain mapping is a relatively new concept for structural health monitoring (SHM), with most methods still in the development stage. Currently, only DIC has seen tentative real-world applications, while other methods remain in lab-based research. Consequently, no specific SHM applications are provided, as there are no other methods that have been used under similar conditions for comparison with DIC.
We also acknowledge the challenges of using optical methods for real-world SHM applications, particularly in scenarios where visual, optical, or physical access to the structure is limited—common in SHM for large-scale or long-distance structures. In response, we have added a paragraph discussing these challenges and proposing the use of hybrid methods to address real-world SHM problems. These changes can be found from 402 to 408 on page 17 and lines 447 to 456 on page 18.
Comments 3: This paper gives an overview of strain mapping methods but does not provide a comprehensive comparison. Several examples were given comparing S4 and ML with FE simulation and DIC results. What about the other methods? As a minimum, a table should be added to summarise and compare the mentioned methods?
Response 3: Thank you for your valuable suggestion. We agree that a comprehensive comparison table would enhance the paper. In response, we have added a performance comparison table that summarizes and compares the key characteristics of the mentioned strain mapping methods. This provides readers with a more straightforward comparison of the advantages and limitations of each technique. The new table can be found on page 19.
Round 2
Reviewer 4 Report
Comments and Suggestions for Authors
The remarks have been addressed.